# Acute Kidney Injury Secondary to Abdominal Compartment Syndrome: Biomarkers, Pressure Variability, and Clinical Outcomes

**DOI:** 10.3390/medicina61030383

**Published:** 2025-02-22

**Authors:** Harun Muğlu, Eslem İnan Kahraman, Erdem Sünger, Ahmet Murt, Ahmet Bilici, Numan Görgülü

**Affiliations:** 1Department of Medical Oncology, Medipol University Faculty of Medicine, Istanbul 34214, Turkey; erdemsunger@gmail.com (E.S.); ahmetknower@gmail.com (A.B.); 2Department of Internal Medicine, Bagcilar Training and Research Hospital, Istanbul 34200, Turkey; eslem.inan.kahraman@outlook.com; 3Department of Nephrology, Cerrahpasa Medical Faculty, Istanbul University-Cerrahpasa, Istanbul 34098, Turkey; ahmet.murt@iuc.edu.tr; 4Department of Nephrology, Bagcilar Training and Research Hospital, Istanbul 34200, Turkey; numangorgulu@gmail.com

**Keywords:** acute kidney injury, abdominal compartment syndrome, intra-abdominal pressure, KIM-1, NGAL, mortality

## Abstract

*Background and Objectives*: Abdominal compartment syndrome (ACS) is a severe clinical condition caused by intra-abdominal hypertension (IAH), often observed in surgical and trauma patients. However, ACS can also develop in non-surgical patients with massive ascites, leading to acute kidney injury (AKI) due to renal hypoperfusion. This study investigates the association between intra-abdominal pressure (IAP) changes, renal biomarkers, and mortality in patients with ACS-related AKI. *Materials and Methods:* A prospective cohort study was conducted on 24 hospitalized patients with ascites due to malignancy, cirrhosis, or heart failure. IAP was measured via the trans-vesical method on the first and seventh days of hospitalization. Serum and urinary biomarkers, including kidney injury molecule-1 (KIM-1), neutrophil gelatinase-associated lipocalin (NGAL), and interleukin-6 (IL-6), were assessed for their correlation with IAP changes. The primary outcome was in-hospital mortality, and the secondary outcomes included AKI progression and the effect of paracentesis on IAP reduction. *Results:* The overall in-hospital mortality rate was 50%. Patients who survived had significantly lower IAP on the seventh day compared to those who died (14.9 ± 3.5 mmHg vs. 20.2 ± 5.6 mmHg, *p* = 0.01). A 25% reduction in IAP was associated with improved kidney function and increased survival (*p* < 0.001). Urinary KIM-1 and serum NGAL levels showed a moderate correlation with IAP (r = 0.55, *p* = 0.02 and r = 0.61, *p* = 0.018, respectively), while IL-6 levels were significantly higher in non-survivors (*p* = 0.03). Paracentesis was associated with improved survival outcomes (*p* = 0.04). *Conclusions:* ACS is a critical but often overlooked cause of AKI in non-surgical patients with massive ascites. Lowering IAP significantly improves renal function and reduces mortality. Urinary KIM-1 and serum NGAL may serve as useful biomarkers for monitoring IAP changes. The early identification and management of IAH through timely interventions such as paracentesis and volume control strategies could improve patient outcomes.

## 1. Introduction

The abdominal cavity maintains a physiological pressure gradient due to the presence of serous fluid between the visceral and parietal layers of the peritoneum. This fluid, typically ranging from 5 to 20 mL, generates a normal intra-abdominal pressure (IAP) of approximately 5–7 mmHg under physiological conditions [1]. This pressure plays a crucial role in maintaining organ function by facilitating the lubrication of abdominal organs and supporting electrolyte diffusion within the peritoneal cavity. However, when IAP exceeds 12 mmHg, a condition known as intra-abdominal hypertension (IAH) develops, which can have significant pathophysiological consequences [2].

IAH is classified into four grades based on IAP measurements: grade 1 (12–15 mmHg), grade 2 (16–20 mmHg), grade 3 (21–25 mmHg), and grade 4 (>25 mmHg). Grades 3 and 4 are typically associated with abdominal compartment syndrome (ACS), a critical condition characterized by progressive organ dysfunction due to sustained elevation of intra-abdominal pressure [3]. ACS is commonly observed in postoperative and trauma patients; however, non-surgical conditions such as massive ascites secondary to cirrhosis, malignancies, or heart failure may also lead to ACS [4].

IAH can manifest in acute, subacute, or chronic forms, depending on the underlying etiology. Acute IAH typically develops within hours and is often linked to trauma, surgical interventions, or internal hemorrhage, where a rapid increase in intra-abdominal volume leads to a sudden elevation in IAP. In such cases, mortality rates can reach up to 60% [5]. Conversely, subacute IAH is more frequently associated with conditions such as decompensated heart failure, advanced liver cirrhosis, or malignancy-related ascites, where IAP rises more gradually. Chronic IAH, on the other hand, is observed in conditions such as pregnancy and morbid obesity, where the risk of ACS is lower but persistent elevation in IAP may contribute to chronic organ dysfunction, including renal impairment [6].

Beyond its direct effects on abdominal organs, IAH has systemic implications. Increased intra-abdominal pressure can elevate intrathoracic pressure, leading to reduced functional residual capacity of the lungs and impaired respiratory mechanics [7]. Similarly, cardiovascular function may be compromised due to decreased venous return, reduced cardiac output, and inferior vena cava compression, predisposing patients to deep vein thrombosis [8]. Furthermore, visceral hypoperfusion can lead to bacterial translocation and sepsis-like syndromes. Acute kidney injury (AKI) is one of the most severe consequences of IAH, resulting from decreased renal perfusion and filtration pressure. When left untreated, IAH-induced AKI may progress to multiple organ dysfunction syndrome (MODS), a critical condition associated with poor prognosis and high mortality [9].

This study aims to investigate the impact of intra-abdominal pressure changes on mortality in patients who develop ACS due to non-surgical medical conditions. In addition to monitoring IAP, we evaluate the prognostic significance of key serum and urinary biomarkers, including interleukin-6 (IL-6), superoxide dismutase (SOD), neutrophil gelatinase-associated lipocalin (NGAL), fibroblast growth factor-23 (FGF-23), and kidney injury molecule-1 (KIM-1), to explore their association with IAP fluctuations and patient outcomes. By elucidating the pathophysiological mechanisms linking ACS and AKI, our findings may contribute to the development of targeted therapeutic strategies for improving prognosis in high-risk patients.

## **2.** Materials and Methods

### 2.1. Study Design and Setting

This prospective observational cohort study was conducted at a tertiary care training and research hospital to assess the impact of IAP changes on mortality in patients who developed AKI secondary to non-surgical medical conditions. The study population consisted of hospitalized patients in the internal medicine ward who had ascites due to malignancy, cirrhosis, or heart failure, and subsequently developed AKI. Ethical approval was obtained from the institutional review board (Approval No: 2018.06.1.05.055), and all participants provided written informed consent before enrollment. A total of 24 patients were included, all of whom completed the study with full follow-up and biomarker assessments. A complete-case approach was applied, as no missing data were present in the dataset. In cases where missing values had occurred, multiple imputation or listwise deletion would have been considered based on the proportion and pattern of missing data. Patients were categorized into survivors and non-survivors, with survival defined as remaining alive until hospital discharge.

### 2.2. Patient Selection; Inclusion and Exclusion Criteria

Patients were included based on the Kidney Disease: Improving Global Outcomes (KDIGO) criteria for AKI, which define AKI as an absolute increase in serum creatinine by ≥0.3 mg/dL within 48 h, an increase in serum creatinine to ≥1.5 times the baseline level within the last seven days, or a urine output of <0.5 mL/kg/hour for six consecutive hours. Additionally, patients were required to be aged 18 years or older and to provide written informed consent prior to study enrollment. Only hospitalized patients with a documented diagnosis of AKI based on the KDIGO criteria were included. The study population was limited to individuals with sufficient cognitive capacity to provide informed consent or, in cases where this was not possible, those with legally authorized representatives who consented on their behalf.

Exclusion criteria encompassed patients with AKI caused by nephrotoxic agents, glomerulonephritis, vasculitis, or other identifiable renal pathologies, as well as those with a history of prior abdominal surgery that could interfere with IAP measurements. Additionally, patients with hepatorenal syndrome, due to its distinct pathophysiology and impact on kidney function, and those with rhabdomyolysis, a separate mechanism of AKI linked to myoglobinuria, were excluded. Recent intravenous contrast administration within the past two weeks was also considered an exclusion criterion, given the potential for contrast-induced nephropathy to confound AKI diagnosis. After applying these criteria, only patients with an IAP ≥ 12 mmHg at admission were included in the final analysis.

Given the underlying renal dysfunction in this patient population, some individuals developed severe kidney failure requiring hemodialysis. The decision to initiate hemodialysis was based on standard clinical indications, including refractory fluid overload, severe metabolic acidosis, electrolyte imbalances, or uremic symptoms. Since dialysis was a direct consequence of AKI progression rather than an independent variable, excluding these patients would have introduced selection bias and reduced the study’s generalizability. Therefore, dialysis-dependent patients were retained in the cohort to ensure comprehensive data representation.

While hemodialysis is known to contribute to systemic inflammation, the primary focus of this study was to assess the impact of IAP reduction on renal function and clinical outcomes. As dialysis was performed based on medical necessity in patients with severe volume overload, its potential inflammatory effects were considered part of the natural course of disease progression rather than a confounding factor. Future studies may further explore the specific impact of dialysis on inflammatory markers through subgroup analyses.

### 2.3. Intra-Abdominal Pressure Measurement

IAP was measured twice, on day 1 and day 7 of hospitalization, using the transvesical method with a Foley catheter [8]. The patient was placed in the supine position, and external abdominal tension was carefully avoided. An amount of 25 mL of isotonic saline was instilled into the bladder, and the catheter was clamped at the intersection of the spina iliaca and midaxillary line. The pressure was then measured in cmH_2_O once fluid fluctuations ceased. To convert values to mmHg, the following correction factor was applied:IAP (mmHg) = IAP (cmH_2_O) × 0.74 

### 2.4. Laboratory Measurements and Clinical Follow-Up

Routine laboratory parameters, including serum creatinine, electrolytes, complete blood count (CBC), albumin, inflammatory markers (C-reactive protein [CRP], procalcitonin), uric acid, bicarbonate (HCO_3_), ferritin, lactate dehydrogenase (LDH), and transaminases, were monitored daily.

Patients were initially treated with diuretics, with dosages adjusted according to volume status and urine output. Patients who were unresponsive to diuretics underwent paracentesis and/or hemodialysis. Data on daily diuresis, total diuretic dosage, and frequency of hemodialysis or paracentesis were systematically recorded.

Initially, patients were treated with diuretics and ultrafiltration (hemodialysis) based on their volume status and renal function. However, patients who did not exhibit clinical improvement in renal function or hemodynamic stability underwent large-volume paracentesis. Notably, all patients included in the paracentesis group were refractory to diuretics and ultrafiltration, suggesting that third-space fluid accumulation limited the efficacy of intravascular volume-depleting therapies [10,11]. This observation aligns with previous studies highlighting the differential effects of diuretics and paracentesis in managing ascites-related intra-abdominal hypertension, where diuretics primarily reduce intravascular volume while paracentesis directly removes extravascular fluid from the peritoneal cavity.

The decision-making process for paracentesis in patients with intra-abdominal hypertension and abdominal compartment syndrome is summarized in Figure 1. This algorithm outlines the appropriate thresholds for intervention, the recommended volume of fluid removal, and the monitoring steps to ensure patient safety.

### 2.5. Biomarker Analysis

To investigate potential prognostic biomarkers associated with IAP changes and mortality, the following markers were measured: SOD, a marker of oxidative stress and ischemia/reperfusion injury, known to fluctuate during AKI; FGF-23, a phosphaturic hormone associated with chronic kidney disease (CKD) progression and potentially indicative of AKI-to-CKD transition; KIM-1, a proximal tubular injury marker that is highly sensitive to ischemia, hypoxia, and toxic injury, and is excreted in urine; NGAL, which is released from tubular cells in response to acute kidney inflammation and is detectable in both serum and urine; and IL-6, a pro-inflammatory cytokine elevated in septic and ischemic AKI, known to drive CRP synthesis [12,13,14].

### 2.6. Biomarker Sample Collection and Analysis

Blood and urine samples were collected on days 1 and 7, coinciding IAP measurements. The timing of biomarker sample collection was strategically designed to reflect specific clinical stages:

Day 1: This represents samples collected immediately after hospital admission. The purpose of this time point was to assess the patients’ acute condition and establish baseline biomarker levels. Early-phase biomarkers, such as serum NGAL and urinary KIM-1, are known to rise rapidly following renal insult, often peaking within the first 24 h [15]. Thus, measurement on day 1 allows for the assessment of the initial injury response.

Day 7: This time point was selected to evaluate the clinical course and treatment response of the patients. It was particularly chosen to observe dynamic changes in biomarker levels in relation to patient stabilization or disease progression. IL-6 and other inflammatory markers may exhibit a more prolonged elevation, reflecting ongoing immune activation and tissue repair processes. Previous studies have demonstrated that serial biomarker assessments over the first week provide valuable prognostic insights, with day 7 representing a critical time point where persistent elevation may indicate prolonged injury or incomplete recovery [16,17,18].

The selection of these two time points aims to monitor changes in biomarker levels during the early and intermediate phases of the disease, providing insights into prognosis and treatment response. By systematically defining these time intervals, our study seeks to understand how clinical variables influence biomarker profiles and their potential role in predicting patient outcomes.

To assess the relationship between IAP changes and renal biomarkers (NGAL, KIM-1, IL-6), correlation analyses were performed using Pearson’s correlation coefficient for normally distributed data and Spearman’s rank correlation for non-normally distributed data.

Samples were stored at −80 °C until analysis. Before testing, samples were gradually thawed (first to −20 °C, then to +4 °C). Biomarkers were measured using commercial ELISA kits with microplate reader RT 2100C and microplate washer RT 2600C devices, manufactured by Rayto Life and Analytical Sciences Co., Ltd., located in Shenzhen, China.

### 2.7. Pharmacological Treatment and Patient Management

Patients included in the study had their pre-existing medications adjusted based on renal dysfunction and clinical necessity before enrollment. Specifically, antihypertensive agents such as angiotensin-converting enzyme (ACE) inhibitors and angiotensin II receptor blockers (ARBs) were discontinued due to their potential impact on kidney function. Similarly, oral antidiabetic drugs were withdrawn, and insulin therapy was initiated when necessary.

For immobile patients, renal dose-adjusted low-molecular-weight heparin was administered for thromboprophylaxis. Diuretic management included furosemide, and amlodipine was used as the primary antihypertensive agent.

Patients with active infections were excluded from the study. Those with elevated C-reactive protein (CRP) and procalcitonin levels underwent a thorough evaluation for potential infectious foci. Urine and blood cultures showed no microbial growth in any patient, and in all cases, the observed inflammatory marker elevations were attributed to the primary disease rather than an active infection.

Given the potential for pharmacological interactions, adjustments to medication regimens were carefully made before study inclusion. Medications known to influence renal function and hemodynamics, such as ACE inhibitors, ARBs, and oral antidiabetic agents, were discontinued as per clinical necessity. No significant drug interactions were observed that could confuse the study results.

### 2.8. Statistical Analysis

Statistical analyses were performed using SPSS version 23.0 (SPSS Inc., Chicago, IL, USA), and a *p*-value < 0.05 was considered statistically significant. The normality of continuous variables was assessed using the Shapiro–Wilk test. Normally distributed continuous variables were reported as mean ± standard deviation (SD) and compared using Student’s *t*-test, while non-normally distributed variables were expressed as median [interquartile range (IQR)] and analyzed using the Mann–Whitney U test. Categorical variables were presented as counts (*n*) and percentages (%) and compared using the Chi-square (χ^2^) test or Fisher’s exact test when expected frequencies were <5.

A complete-case approach was applied, as no missing data were present in the dataset. In cases where missing values had occurred, multiple imputation or listwise deletion would have been considered based on the proportion and pattern of missing data. Patients were categorized into survivors and non-survivors, with survival defined as remaining alive until hospital discharge.

Pearson’s correlation coefficient was used for normally distributed variables, while Spearman’s rank correlation coefficient was applied for non-normally distributed variables to evaluate the relationship between IAP changes and biomarker levels (NGAL, KIM-1, IL-6).

A multivariate logistic regression model was constructed to determine independent predictors of mortality and AKI progression. The model incorporated baseline and day 7 IAP, a ≥25% reduction in IAP (binary), NGAL and KIM-1 levels, IL-6 levels, serum creatinine, bicarbonate (HCO_3_), paracentesis, age, and sex as independent variables. Regression results were presented as odds ratios (OR) with 95% confidence intervals (CI), and model fit was assessed using the Hosmer–Lemeshow test.

Paracentesis was performed based on clinical necessity as part of the patient management approach. Therefore, comparisons were made based on the degree of IAP reduction rather than the presence or absence of the procedure. A sensitivity analysis was conducted to account for potential confounding effects of underlying malignancy or heart failure by stratifying patients according to their primary diagnosis.

## 3. Results

A total of 24 patients were included in the study, with a mean age of 69.2 ± 9.0 years, and a median hospital stay of 14.8 ± 7.1 days. The primary underlying conditions were congestive heart failure (*n* = 12), cirrhosis (*n* = 6), malignancies (*n* = 6; ovarian cancer (*n* = 2), colon cancer (*n* = 2), endometrial cancer (*n* = 1), and gastric cancer (*n* = 1))**.** The overall in-hospital mortality rate among patients with IAH was 50%**.**

IAP levels did not significantly differ between survivors and non-survivors (22.2 ± 5.7 mmHg vs. 23.1 ± 4.1 mmHg, *p* = 0.34). However, at day 7, patients who survived had significantly lower IAP compared to those who died (14.9 ± 3.5 mmHg vs. 20.2 ± 5.6 mmHg, *p* = 0.01). A reduction of at least 25% in IAP was significantly associated with AKI recovery (*p* = 0.01) and overall survival (*p* < 0.001). Logistic regression analysis demonstrated that changes in IAP were independently associated with mortality (OR: 1.88, 95% CI: 1.19–2.95, *p* = 0.006). Additionally, paracentesis was found to be significantly associated with improved survival outcomes (*p* = 0.04), whereas the use of diuretics and hemodialysis did not differ between groups.

Among the biomarkers studied, urinary KIM-1 and serum NGAL levels showed a significant correlation with IAP levels (r = 0.55, *p* = 0.02 and r = 0.61, *p* = 0.018, respectively). Furthermore, IL-6 levels were significantly higher in non-survivors compared to survivors (11.4 ± 7.0 pg/mL vs. 5.6 ± 5.5 pg/mL, *p* = 0.03)**.** However, no statistically significant difference or correlation was found for SOD or FGF-23 with IAP levels or clinical outcomes. A detailed comparison of the groups and associated variables is presented in Table 1.

These findings highlight the clinical significance of IAP reduction in improving renal function and overall survival. Among the studied biomarkers, urinary KIM-1 and serum NGAL emerged as the most reliable indicators of IAP changes and renal recovery, demonstrating strong correlations with disease progression. These results suggest that biomarker-based monitoring may offer a valuable non-invasive alternative to guide therapeutic interventions and improve patient outcomes in ACS-related AKI.

## 4. Discussion

IAH is a pathological condition characterized by elevated IAP, which can progress to ACS. ACS is defined by a sustained IAP exceeding 20 mmHg, leading to organ dysfunction or failure. Early recognition and timely intervention to reduce IAP are crucial to prevent organ injury and improve clinical outcomes. In ACS, increased IAP impairs renal venous return and reduces cardiac output, contributing to the development of AKI and exacerbating systemic hemodynamic instability. Management strategies include regular monitoring of IAP, optimizing fluid balance, and, when necessary, surgical decompression. Prompt diagnosis and appropriate treatment are essential to mitigate the high morbidity and mortality associated with ACS [19,20,21].

This study specifically focused on non-surgical patients with ACS-related AKI and employed intravesical pressure measurements as a surrogate for IAP monitoring, ensuring the exclusion of patients with prior abdominal surgeries to enhance measurement accuracy.

Elevated IAP has been strongly associated with impaired renal function and increased mortality in critically ill patients. Studies have demonstrated that IAH can lead to AKI and adversely affect patient outcomes [22,23,24].

In a study by Demarchi et al., AKI developed when IAP exceeded 8 mmHg in a cohort of 60 patients. This finding underscores the sensitivity of renal function to even modest increases in IAP. Additionally, increased IAP has been linked to higher mortality rates in critically ill patients, particularly in those undergoing surgery or suffering from sepsis, acute pancreatitis, or trauma [25].

In pediatric intensive care unit (ICU) settings, even a 1 mmHg increase in IAP was shown to negatively affect prognosis and prolong ICU stay. Similarly, a study of postoperative ICU patients demonstrated a significant association between elevated IAP and mortality, with rates increasing from 10 to 25% in grade 1 and 2 IAH to 75–90% in ACS cases [26,27].

These findings highlight the importance of regular monitoring of IAP in critically ill patients. Early detection and intervention are crucial to prevent the progression of IAH to ACS and to mitigate associated risks such as AKI and increased mortality. Management strategies may include optimizing fluid balance, using diuretics judiciously, and considering surgical decompression when necessary. Implementing these measures can improve patient outcomes and reduce the incidence of IAP-related complications.

To the best of our knowledge, no previous study has systematically investigated ACS secondary to malignancy, heart failure, or liver cirrhosis, nor assessed the impact of IAP changes on renal function and survival in this patient population. The majority of existing literature has focused on surgical ICU patients who developed ACS as a result of sepsis and/or trauma. In contrast, the present study included only non-ICU patients, all of whom had massive ascites and were managed with diuretic therapy, hemodialysis/hemofiltration, and/or paracentesis. The observed in-hospital mortality rate of 50% in our study is consistent with previously reported mortality rates ranging between 40% and 100% in ACS patients [28].

Our findings highlight the clinical significance of reducing IAP in improving renal function and overall survival. A ≥25% reduction in IAP was independently associated with improved kidney function and reduced mortality, suggesting that IAP modulation should be a key therapeutic target in ACS management.

In addition to conventional IAP monitoring via trans-vesical catheterization, ultrasound-based approaches have been explored as non-invasive methods to estimate intra-abdominal fluid volume and pressure. Several studies suggest that ultrasound can detect even minimal ascitic fluid (as low as 100 mL) and may provide indirect markers of IAH, such as the inferior vena cava collapsibility index and diaphragmatic excursion [10]. Although ultrasound cannot directly measure IAP, it may serve as a complementary tool, particularly in patients with contraindications to invasive monitoring. Future studies should investigate the correlation between ultrasound-based parameters and IAP trends to refine diagnostic and therapeutic strategies in ACS management [8].

Based on these findings, a structured approach to IAH management is critical to preventing AKI progression. In addition to therapeutic paracentesis, optimizing fluid balance and monitoring biomarkers (NGAL, KIM-1) should be integrated into clinical practice. Figure 2 illustrates a practical algorithm outlining stepwise actions based on IAP severity to mitigate AKI risk and guide early intervention.

Our findings emphasize the potential superiority of paracentesis over diuretics and ultrafiltration in managing ACS-related AKI in patients with massive ascites. While diuretics and ultrafiltration aim to mobilize intravascular fluid, their effectiveness is often limited in cases of severe ascites due to the extravascular nature of third-space fluid accumulation. This discrepancy may explain the lack of clinical improvement observed in patients receiving diuretics and ultrafiltration alone. In contrast, large-volume paracentesis directly decreases intra-abdominal pressure, thereby improving renal perfusion and reducing the risk of organ dysfunction. These results are in agreement with previous studies demonstrating that paracentesis is the preferred intervention in patients with refractory ascites and diuretic resistance [29,30].

Given that ACS can precipitate AKI in patients with massive ascites, paracentesis emerges as a reasonable intervention to rapidly alleviate intra-abdominal pressure [31]. Additionally, diuretic therapy and ultrafiltration via hemodialysis may further enhance renal function and reduce mortality risk in these patients. To optimize patient outcomes, clinicians should implement early and effective volume control strategies to prevent progression to multi-organ dysfunction [32].

AKI remains a critical concern in critically ill patients, particularly those with IAH and ACS, where early detection is essential for timely intervention. Traditional diagnostic markers, such as serum creatinine and blood urea nitrogen, often exhibit delayed elevation following kidney injury, limiting their utility in early diagnosis. In this context, novel biomarkers like NGAL and KIM-1 have emerged as promising tools for detecting renal dysfunction at earlier stages [33,34,35].

For patients in whom abdominal hypertension is suspected, regular IAP monitoring can guide therapeutic decisions. Although dependent on operator expertise, the trans-vesical method remains a simple and practical approach for IAP measurement. However, biomarkers may serve as alternative non-invasive monitoring tools. In this study, urinary KIM-1 and serum NGAL emerged as promising biomarkers correlated with IAP, suggesting their potential roles in assessing treatment response.

Since NGAL levels can be influenced by extrarenal factors such as systemic inflammation, chronic kidney disease, and other non-renal conditions, we applied several measures to improve its specificity as a biomarker for AKI. To minimize confounding, patients with active infections and systemic inflammatory disorders were excluded. Additionally, we analyzed NGAL trends over time rather than relying on single measurements, allowing us to distinguish transient systemic effects from renal-specific elevations. NGAL levels were also evaluated alongside other kidney injury markers, such as KIM-1 and serum creatinine, to enhance diagnostic accuracy. These measures align with previous studies indicating that NGAL rises early and transiently in response to ischemic or nephrotoxic kidney injury, unlike chronic or inflammatory conditions, which cause prolonged elevations. While our findings support the use of NGAL as an early biomarker for AKI in ACS patients, its potential limitations in the presence of systemic pathology should be considered.

Despite the growing body of literature on NGAL and KIM-1, no previous studies have specifically examined these biomarkers in ACS patients. Given that ACS is associated with increased IAP, which can lead to renal dysfunction due to compromised renal perfusion, investigating NGAL and KIM-1 in this population is of significant clinical importance. Our findings support the use of NGAL as an early biomarker for AKI in ACS patients, but its potential limitations in the presence of systemic pathology should be considered. Additionally, serum IL-6 levels were significantly higher in non-survivors, indicating a possible prognostic value for ACS patients. IL-6 levels may therefore help identify high-risk patients who require more aggressive intervention strategies.

Despite its strengths, this study has certain limitations. Firstly, the small sample size may have limited statistical power, as the study focused specifically on patients with ACS rather than including those with milder forms of grade 1 or 2 IAH. Future studies with larger cohorts may help refine these findings by incorporating a broader range of IAP levels. Secondly, although malignancy is an independent risk factor for mortality, the proportion of malignancy cases did not significantly differ between survivors and non-survivors, suggesting that IAP changes were a more critical determinant of outcomes in this study. Thirdly, due to the limited sample size, subgroup analyses for different ACS etiologies (heart failure, cirrhosis, and malignancy) could not be performed, necessitating further research to explore potential differences in disease progression and response to treatment.

In summary, these findings highlight the importance of early IAP monitoring and intervention in patients with ACS-related AKI. Lowering IAP through paracentesis, diuretics, and hemodialysis can improve renal function and overall survival. Moreover, biomarkers such as KIM-1, NGAL, and IL-6 may serve as valuable tools for risk stratification and treatment optimization. Further prospective studies with larger cohorts are warranted to validate these findings and refine management strategies for this high-risk patient population.

## 5. Conclusions

ACS is a critical condition characterized by sustained IAH, leading to organ dysfunction, including AKI. While ACS is well-documented in surgical and critically ill patients, its impact in non-surgical settings, particularly in patients with massive ascites due to heart failure, cirrhosis, or malignancy, remains underrecognized.

Our study demonstrates that reducing IAP significantly improves renal function and reduces in-hospital mortality in ACS-related AKI. A reduction of at least 25% in IAP was strongly associated with better kidney function and survival outcomes. Additionally, urinary KIM-1 and serum NGAL levels showed significant correlations with IAP changes, suggesting their potential as non-invasive biomarkers for monitoring disease progression and treatment response.

The findings highlight the importance of early recognition and management of ACS in non-surgical patients. Timely interventions, such as paracentesis, can effectively lower IAP and improve clinical outcomes, particularly in patients who do not respond to diuretics. Integrating biomarker-based monitoring into clinical practice may further enhance patient management by providing early indications of disease progression.

Future studies should focus on larger patient cohorts to validate these findings and explore alternative therapeutic strategies. Investigating the role of pharmacological interventions and non-invasive monitoring techniques, such as ultrasound-based IAP estimation, could further optimize ACS management.

In conclusion, our study underscores the need for early detection and targeted intervention in ACS-related AKI. Lowering IAP through appropriate therapeutic strategies can improve renal function and survival, while biomarker-based monitoring may offer valuable guidance for clinical decision-making. A multidisciplinary approach integrating IAP reduction, fluid management, and biomarker assessment may significantly enhance outcomes in high-risk patients.

## Figures and Tables

**Figure 1 medicina-61-00383-f001:**
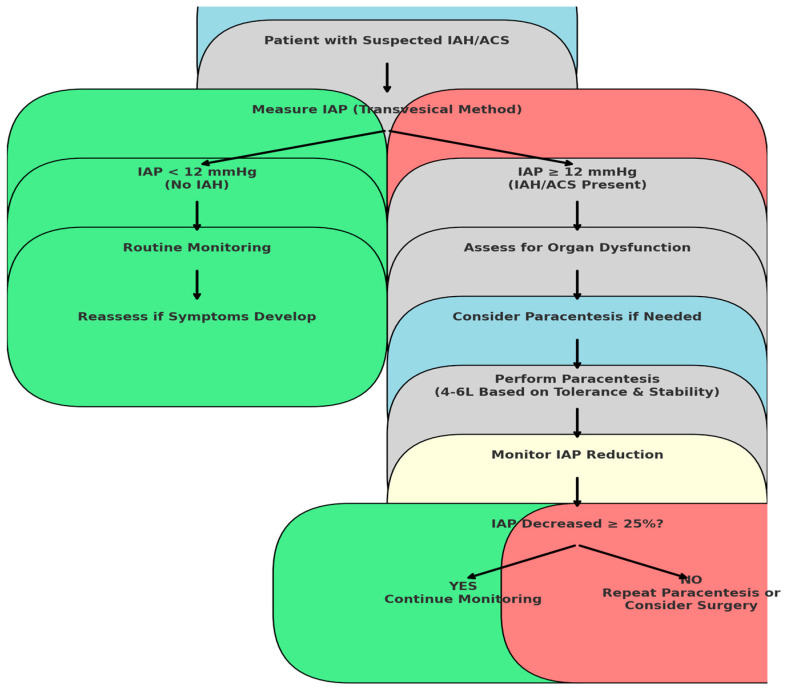
Algorithm for paracentesis in intra-abdominal hypertension and abdominal compartment syndrome. The algorithm guides decision-making based on intra-abdominal pressure levels, clinical stability, and recommended paracentesis volumes (4–6 L based on tolerance and hemodynamic stability).

**Figure 2 medicina-61-00383-f002:**
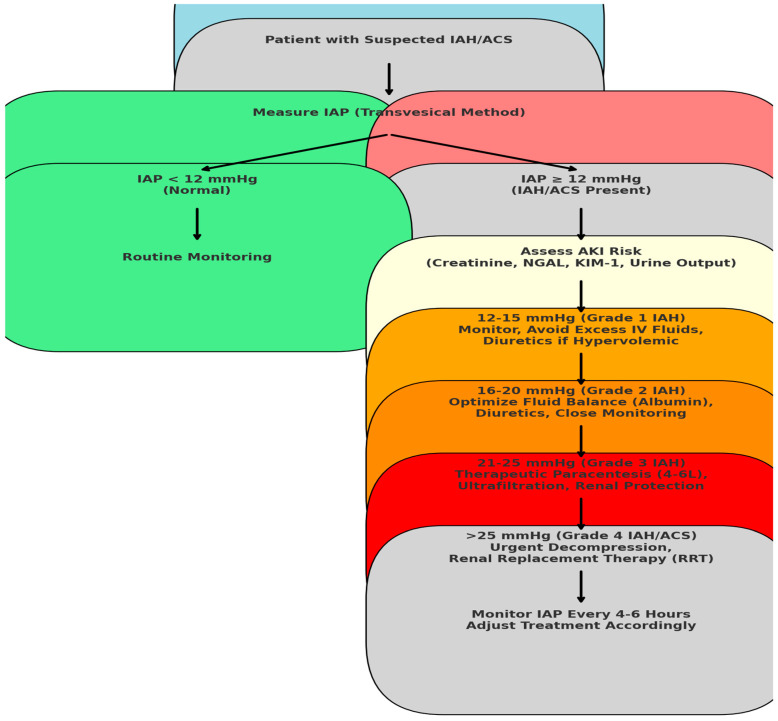
Algorithm for preventing AKI progression in IAH/ACS. This algorithm provides a stepwise approach based on intra-abdominal pressure levels, integrating paracentesis, diuretic use, fluid balance strategies, and renal replacement therapy for high-risk patients.

**Table 1 medicina-61-00383-t001:** Comparison of patients survivors and non-survivors in their hospital admissions.

Parameter	Survivors (*n* = 12)	Non-Survivors (*n* = 12)	*p*-Value
Age (years)	66.5 ± 9.3	72.1 ± 8.1	0.1
Male (*n*, %)	3 (25%)	3 (25%)	1.0
Malignancy (*n*, %)	4 (33.3%)	2 (16.6%)	0.34
Creatinine (mg/dL)	2.6 [2.0–4.57]	3.0 [1.7–4.7]	0.2
Sodium (mmol/L)	134.25 ± 2.9	131.3 ± 5.0	0.09
Potassium (mmol/L)	4.2 ± 1.2	4.4 ± 0.8	0.3
Phosphorus (mg/dL)	4.2 ± 1.1	5.4 ± 1.4	0.029
Calcium (mg/dL)	7.9 ± 1.6	8.1 ± 0.6	0.4
Uric acid (mg/dL)	8.3 ± 3.0	12.3 ± 5.4	0.03
Albumin (g/dL)	2.8 ± 0.4	2.4 ± 0.3	0.05
^c^ CRP (mg/L)	43.5 ± 65.8	128.9 ± 71.4	0.01
Procalcitonin (ng/mL)	0.6 ± 0.7	3.0 ± 1.7	0.001
^d^ HCO_3_ (mEq/L)	22.5 ± 8.4	15.8 ± 4.9	0.02
^a^ ALT (U/L)	32.0 ± 9.5	139.9 ± 320.3	0.26
^‡^ AST (U/L)	30.2 ± 36.7	177.8 ± 357.4	0.16
^b^ LDH (U/L)	363.5 ± 160.3	678.9 ± 599.6	0.09
Hemoglobin (g/dL)	10.1 ± 1.1	10.0 ± 1.6	0.86
Ferritin (ng/mL)	241.4 ± 217.5	503.1 ± 598.5	0.16
Daily diuresis (mL)	1366 ± 1117	1108 ± 631.6	0.49
Diuretic dose (mg)	211.6 ± 150.3	160.8 ± 81.6	0.26
Need for hemodialysis (*n*, %)	6 (50%)	5 (41.6%)	0.68
Paracentesis (*n*, %)	8 (66.6%)	3 (25%)	0.04
IAP on first day (mmHg)	23.1 ± 4.1	22.2 ± 5.7	0.34
25% decrease in IAP (*n*)	12 (100%)	1 (8.3%)	0.0001
IAP on seventh day (mmHg)	14.9 ± 3.5	20.2 ± 5.6	0.01
* Change in IAP * between first and seventh day (mmHg)	9 ± 3.9	1.08 ± 2.8	0.001
Follow-up time (days)	14.0 ± 4.6	12.1 ± 2.8	0.21
^§^ IL-6 levels (pg/mL)	5.6 ± 5.5	11.4 ± 7.0	0.03

* IAP: Intraabdominal pressure; ^§^ IL-6: Interleukin-6; ^a^ ALT: Alanine aminotransferase; ^‡^ AST: Aspartate aminotransferase; ^b^ LDH: Lactate dehydrogenase; ^c^ CRP: C-reactive protein; ^d^ HCO_3_: Serum bicarbonate.

## Data Availability

The data of this study is available from the corresponding author upon a reasonable request.

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
