# Peer review of "Acute Kidney Injury Secondary to Abdominal Compartment Syndrome: Biomarkers, Pressure Variability, and Clinical Outcomes"

_medicina, 2025, doi:10.3390/medicina61030383_

Round 1
Reviewer 1 Report
Comments and Suggestions for Authors
Dear Authors,
Thank you for presenting an intriguing and practically significant article. I have a few questions and suggestions that I believe could further enhance your work:
-
Could you please clarify the Biomarker Sample Collection and Analysis: Blood and urine samples were collected on Days 1 and 7. Could you specify what these days represent? Is Day 1 the day of hospital admission, the onset of illness, or another specific point in time?
-
The study included 24 patients. Could you clarify how many patients enrolled the study and whether all of these patients completed the study?
-
How did you account for the influence of extrarenal pathology on NGAL levels when interpreting the results?
-
You claim that paracentesis significantly improves the prognosis. However, in the discussion, you mention the effects of diuretics and ultrafiltration. Were diuretics and ultrafiltration used in your patients? If so, is it possible to compare the effectiveness of ultrafiltration and diuretics with paracentesis?
-
Regarding the practical significance of paracentesis, could you recommend an algorithm for what pressure, how much, and in what volume paracentesis should be performed?
-
Is it possible to use ultrasound diagnostics to assess the level of pressure and fluid in the abdominal cavity?
-
I would recommend expanding and updating the literature sources.
-
It would be highly beneficial to include a practical algorithm of actions based on the degree of pressure increase to prevent the progression of acute kidney injury.
Thank you for your attention to these matters.
Kind regards,
Reviewer 2 Report
Comments and Suggestions for Authors
The papers describes an interesting research carried out about the relationship between AKI and an increased IAP. The topic is original and interesting.
some bias, on my opinion, should be underlined.
Patients who underwent hemodialysis and patients in medical treatment alone were not distinguished. Dialysis is per se a further inflammatory trigger due to many reasons. So Authors should consider the hypothesis that Dialyzed patients could be excluded from the study.
Medical treatment should be more carefully listed in order to exclude potential pharmacological interactions.
PTH is not a useful marker in such a clinical condition and the majority of markers are the same employied in routinary AKI treatment. So the original idea lacks its interesting as the pages pass away.
Methods and statistycal tests should be claRIFYIED AND IMPROVED.
Discussion and Conclusions should be improved.
Thank You for Your work!
Round 2
Reviewer 1 Report
Comments and Suggestions for Authors
Dear authors, thank you for your comments, I haven't any more questions.
Reviewer 2 Report
Comments and Suggestions for Authors
No further questions. Thank You